# Clinical practice guidelines for cardiovascular disease: how is depression addressed? Protocol for a systematic review

Dana Blatch Armon [1], Aliki Buhayer [1], Kevin Dobretz [2], Gunther Meinlschmidt [3,4], Edouard Battegay [1,5]

DBA and AB contributed equally.

For numbered affiliations see end of article.

**Correspondence to**
Dr Dana Blatch Armon;
dana.blatch@mail.huji.ac.il

## ABSTRACT

**Introduction** Depression frequently affects patients with cardiovascular disease (CVD). When these conditions co-occur, outcomes such as quality of life and life expectancy worsen. In everyday practice, this specific and prevalent disease-disease interaction complicates patient management. Clinical practice guidelines (CPGs) aim to provide the best available advice for clinical decision-making to improve patient care. This study will aim to evaluate how CPGs specifically address depression in patients with CVD, and whether they provide any operational guidance for screening and management of depression in the primary care and outpatient setting.

**Methods and analysis** We will conduct a systematic review of CPGs on CVD management published from 2012 to 2023. A broad literature search for guidelines will be performed through electronic medical databases, grey literature search tools, and websites of national and professional medical organisations.
Based on the inclusion criteria, two independent reviewers will evaluate eligible guidelines for screening and management recommendations on depression in patients with CVD. Additional points to be evaluated will be any mention of drug–drug or drug–disease interactions, other aspects of specific relevance to treating physicians, as well as general information on mental health. We will assess the quality of CPGs with a recommendation regarding depression in CVD patients using the Appraisal of Guidelines for Research and Evaluation II.

**Ethics and dissemination** As this systematic review is based on available published data, ethics approval and consent are not applicable. Our intent is that our results will be published in a peer-reviewed journal, presented at international scientific meetings, and distributed to healthcare providers.

**PROSPERO registration number** CRD42022384152.

## STRENGTHS AND LIMITATIONS OF THIS STUDY

⇒ The rigorous literature search method we developed aims to produce an extensive record of cardiovascular disease clinical practice guidelines (CPGs), to accurately describe how depression screening and management is addressed in these patients.
⇒ Our literature search will cover the past 10 years to provide insights into current and clinically relevant guidelines, while also facilitating the identification of possible trends in recommendations.
⇒ This systematic review will use the Appraisal of Guidelines for Research and Evaluation II instrument to evaluate the quality of the selected guidelines, including all relevant recommendations.
⇒ This study will include only CPGs published in English, excluding any guidelines produced in other languages.

## INTRODUCTION

Despite substantial advances in the field, cardiovascular disease (CVD) is still the leading cause of morbidity and mortality in the world.[1] Improvements in primary prevention and management of risk factors have lowered the risk of cardiovascular events and reduced age-specific mortality over the past decades.[1 2] However, higher life expectancy has led to a rise in comorbid conditions in CVD patients.[3] Multimorbidity, commonly described as the occurrence of two or more chronic conditions in one individual,[4] is associated with increased mortality, decreased quality of life and increased use of healthcare services,[3 5 6] including visits to general practitioners and other medical specialists.[7] In addition, multimorbidity leads to challenging situations in the treatment and management of these patients, further complicating medical decision-making.[7 8] The relationship between CVD and multimorbidity is well established,[9–11] and among multimorbidities, depression is one of the most common co-occurring conditions.[12]

Depression is a common mental disorder that interferes with a person's mood and behaviour.[13] It affects up to 10% of adults throughout their lifetime[14] and is considered to be the leading cause of disability worldwide.[15] In patients with CVD, the occurrence of depression is estimated to be one in five.[12]

When controlling for traditional CVD risk factors, the presence of depressive symptoms even without a clinical diagnosis of depressive disorder can predict the incidence of coronary artery disease in healthy individuals,[16] as well as secondary events in these patients,[17–20] and adverse outcomes following coronary artery bypass grafting.[21] In patients with CVD, depression was also associated with an elevated risk of failure to adhere to medications and a healthy lifestyle.[22 23] Patients with depression may be less likely to engage in behaviours that lower the risk of future cardiovascular events, such as physical activity, dietary modifications, smoking cessation, stress management and treatments against substance abuse.[24 25]

Inadequate treatment of depression is associated with a higher risk of adverse cardiovascular events.[26] Primary care physicians play an important role in recognising and managing depression since an estimated 60% of mental healthcare delivery occurs in the primary care setting.[27] Despite high evidence on the importance of diagnosis and treatment of depression,[28] implementation in practice remains limited,[29] and over 50% of patients with depression in primary care remain unrecognised and inadequately treated.[30–33]

Clinical practice guidelines (CPGs) provide the best available evidence to support clinical decision-making and aim to improve care and outcomes,[34] however, they vary in their recommendations regarding screening for depression. For example, the US Preventive Services Task Force recommends screening for depression in the general adult population,[35] whereas official guidelines by the National Institute for Health and Clinical Excellence (NICE), as well as the Canadian Task Force on Preventive Health Care recommend not to screen routinely for depression in the primary care setting.[36 37]

A recent state-of-the-art review on depression in patients with CVD showed that guideline recommendations for screening for depression are only partially available and inconsistent.[38] The recommendations tended to focus on specific subpatient groups and lacked a systematic approach, also regarding which screening tools to use, leaving an incomplete picture of practical recommendations for practitioners to follow in their daily practice.[38]

### Evidence gap
The systematic review of existing CPGs is an approach that is increasingly being used in an attempt to characterise the nature of clinical guidance on a topic of interest.[34] To our knowledge, no such review has yet been performed regarding how depression is addressed in CVD guidelines. As standardised screening and treatment pathways offer the potential for early identification and improved management of both CVD and depression in patients treated for specific CV diseases,[39–41] we aim to conduct such a review in order to provide a clear picture of the recommendations currently available and identify gaps in knowledge. This step is crucial to provide guidance for future research and, ultimately, for improving the quality of evidence-based clinical recommendations.

### Aim
The aim of this systematic review will be to identify, appraise and describe how CPGs for CVD management specifically address depression, and whether they provide operational guidance for screening and management to practitioners.

This systematic review is part of a larger review currently in process that is meant to provide a comprehensive picture of knowledge and evidence gathered so far on patients with concurrent CVD and depression. It is also intended to identify knowledge gaps in the current state of clinical guidance, thereby promoting further research to produce evidence-based guidance for future guidelines to further improve treatment for patients.

## OBJECTIVES
1. Identify published CPGs on the management of CVD in the primary care and outpatient setting.
2. Among all qualifying CPGs on CVD, identify those that mention and provide recommendations for depression as comorbidity or multimorbidity.
3. Assess the quality of recommendations using the Appraisal of Guidelines for Research and Evaluation version II (AGREE-II) instrument.
4. Summarise the current situation with regard to the consideration of depression in CPG on CVD.

## METHODS AND ANALYSIS
For the preparation of this protocol, we followed specific methodological guidance for conducting systematic reviews of CPGs, as such reviews require a tailored approach to design and execution compared with other systematic reviews.[34] This protocol follows the Preferred Reporting Items for Systematic Review and Meta-Analysis Protocols (PRISMA-P) guidelines for the reporting of systematic review protocols.[42] The completed checklist for this protocol is available in online supplemental file 1.

### Patient and public involvement
Patients and/or the public were not involved in the design, or conduct, or reporting, or dissemination plans of this research protocol.

### Eligibility criteria
The PICAR[43] framework was used to guide inclusion and exclusion criteria and is described in table 1, as recommended by the PRISMA-P framework for protocols of systematic reviews. These selected guidelines will be assessed for recommendations regarding depression in patients with CVD. We will use the Preferred Reporting Items for Systematic Reviews and Meta-Analyses guidelines (PRISMA) to report the results of this review.[44]

### Search strategy and information sources
To identify all relevant CPGs, the search strategy was developed on the basis of the Johnston paper recommendations for a systematic review of CPGs and under

**Table 1** PICAR statement

| | |
|---|---|
| P: Population, clinical indication(s) and condition(s) | Study population: adults 18 years of age and above with CVD<br>Clinical indication: management of patients with CVD in the primary/specialty outpatient setting<br>Clinical condition: CVD |
| I: Intervention(s) | Any intervention focusing on diagnosis and/or treatment of depression |
| C: Comparator(s) | No comparator. All aspects of depression diagnosis and management will be taken into consideration |
| A: Attributes of eligible CPGs | ► Language: English<br>► Time range: 10 years, starting from search date. CPGs that were published before 2013 and have not been renewed or updated since are considered out of date on the basis of the rapid advances in the field of CVD over the past decade<br>► Publishing region: no restriction<br>► Versions: latest versions only<br>► Developing publishing organisation: endorsed by national or international scientific societies, professional colleges, charitable organisations and government organisations will be included<br>► Publication status: only published CPGs<br>► Development process: explicitly evidence-based and/or consensus-based<br>► System of rating evidence: use of a system to rate the level of evidence within CPGs is not an eligibility criterion; but will be taken into consideration when available and stated clearly<br>► Scope: CPGs for the general management of adult patients with CVD (including heart failure, coronary artery disease, stroke, peripheral artery disease, aortic disease, diabetes, dyslipidaemia and hypertension management) in the community setting |
| R: Recommendations characteristics | Recommendations covering screening, diagnosis, and pharmacological and non-pharmacological management of depression are of interest |

CPGs, clinical practice guidelines; CVD, cardiovascular disease.

the guidance of an experienced bioinformatician (KD). We will conduct a systematic search in the following databases: PUBMED, TRIP, EMBASE, Guidelines International Network and Guideline Central, using search terms and methods adjusted to the specificities of each database (example in online supplemental file 2). In order to locate additional relevant CPGs not available in academic journals, we will extend our search range to include additional grey literature that will target guidelines by specifically searching on the Society Guideline Link on Uptodate, WHO guidelines section, and the websites of national and professional medical organisations (eg, NICE, Scottish Intercollegiate Guidelines Network).

## Guideline selection

All results collected in the literature search will be summarised in a table and submitted to two independent reviewers (DBA and AB) who will conduct a two-step selection procedure for eligibility, based on the predefined inclusion and exclusion criteria:

1. First step: titles and abstracts will be screened for relevance. Duplicates will be removed in this step. Potentially relevant CPGs where an abstract is not provided will be passed to the second screening stage.
2. Second step: full-text screening to assess CPGs for final inclusion in the review.

Records will be excluded if both reviewers agreed they were not eligible. Any disagreements will be resolved

by consensus. If consensus cannot be met between the two reviewers, a third reviewer (EB) will also assess the particular record to determine its eligibility.

## Inclusion criteria

1. CPGs for general outpatient management of adult patients with CVD (including specific guidelines for heart failure, coronary artery disease, stroke, peripheral arterial disease, aortic disease, diabetes, dyslipidaemia and hypertension management) in the community setting.
2. The CPG is published in English.
3. The CPG was published within the past 10 years.
4. The CPG is the most recent version.
5. The full CPG is available online.
6. The CPG was written by national and international academic organisations.

## Exclusion criteria

1. Guidelines limited to a specific aspect of CVD management, such as screening, pharmacological management, specific interventions, acute and hospital setting, or not relevant to primary care or outpatient care.
2. Target population under 18 years of age.

## Data extraction and analysis
### Data management and extraction

Once the final set of included CPGs has been obtained, DBA and AB will conduct a search in the full text for any

mention of depression, using the search terms "depress" "mental" "psych" and "mood". These search terms will be used to identify all the places within the guidelines which mention the topic of mental health, in order to extract any information specifically regarding depression. Additional information regarding other mental disorders will only be evaluated for relevance and briefly summarised and presented, as it is not the focus of this work.

First, all results collected will be summarised in a table containing the CPG title, year of publication, organisation and country, and the exact context in the document (page number, chapter, line, table, or figure number).

Second, DBA and AB will assess and categorise each result for any operational guidance on screening and/ or management of depression in patients with CVD. Any mention of interactions and points of consideration for treating physicians, as well as general information on mental health issues will also be included.

Specifically, the following aspects will be examined and classified:

1. Screening recommendation: population, how to screen (who, what screening tool), when (and how often) and what to do with a positive screening result (treatment, referral).
2. Management: pharmacological and non-pharmacological treatment modalities, drug–drug or drug–disease interactions to consider.

Special consideration will be given to the level of evidence on which the recommendation is based and will be reported when available. In addition, the quality of the selected CPGs will be assessed using the updated AGREE-II instrument.[45] As the study progresses, the datasheet may evolve to fit the study's objectives better. The full list of variables of interest is shown in box 1.

## Results

The systematic review will aim at providing results for the following planned steps:

1. A flowchart of the search strategy.
2. A detailed summary of available recommendations (any type) on depression listed by identified CPG on CVD.
3. Quality assessment sheets of CPGs using the AGREE-II methodology.

## Significance of the study

Patients with CVD have a worse prognosis if they suffer from depression. This study aims to establish what guidance CPGs for CVD provide regarding screening and management of depression. By evaluating the current state of recommendations available and identifying the gaps in knowledge, this study may serve as a guide for tailoring future research in this field. This step is crucial for the development of evidence-based recommendations for future CPGs aiming at providing better guidance on the management of depression in patients with CVD.

## Ethics and dissemination

As this study is a systematic review, a secondary analysis of published data will be performed, and no human participants involved in this research. Therefore, in this situation ethics approval and consent are not applicable.

This study is registered on the International Prospective Register of Systematic Reviews (PROSPERO) as of 26 December 2022 and will be updated as needed and as main results will be published.

The results of this study will be reported in accordance with the PRISMA statement[44] and disseminated through publication in a peer-reviewed journal, presentation at international scientific conferences and contact with medical associations responsible for writing guidelines.

**Author affiliations**
[1]Department of Psychosomatic Medicine, International Center for Multimorbidity and Complexity in Medicine, University of Zurich, University Hospital Basel, Merian Iselin Klinik, Basel, Switzerland
[2]Cardiology, Geneva University Hospitals, Geneva, Switzerland
[3]Department of Digital and Blended Psychosomatics and Psychotherapy, Psychosomatic Medicine, University Hospital Basel and University of Basel, Basel, Switzerland
[4]Division of Clinical Psychology and Cognitive Behavioural Therapy, International Psychoanalytic University Berlin, Berlin, Germany
[5]Psychosomatic Medicine, University Hospital Basel, Basel, Switzerland

**Contributors** EBA, AB and DB were involved in the conceptualisation of the study. EBA, AB, DB and GM designed the protocol and developed the methodology for the literature search strategy. KD conduct the literature search. DB and AB will conduct the data analysis, quality assessment and formal analysis. EBA will supervise the study and is in charge of funding acquisition. All authors have made important intellectual content contributions to the protocol, and reviewed and edited its final version.

**Funding** EBA received a non-restricted educational and research grant of the Zurich Academy of Internal Medicine, https://www.my-zaim.ch. GM received funding from the Stanley Thomas Johnson Stiftung & Gottfried und Julia Bangerter-Rhyner-Stiftung under projects no. PC 28/17 and PC 05/18, from Gesundheitsförderung Schweiz under project no. 18.191/K50001, from the Swiss Heart Foundation under project no. FF21101, from the Research Foundation of the International Psychoanalytic University (IPU) Berlin under projects no. 5087 and 5217, from the Swiss National Science Foundation (SNSF) under project no. 100014_135328, from the Hasler Foundation under project No. 23004, in the context of a Horizon Europe project from the Swiss State Secretariat for Education,

---

**Box 1    List of variables**

⇒ Name of variable.
⇒ CPG topic (CVD or specific CV-related conditions, including CAD, HF, stroke, PAD, aortic disease, hypertension, diabetes, dyslipidaemia).
⇒ CPG title.
⇒ Organisation/society responsible for the guideline.
⇒ Year of publication.
⇒ Source of funding.
⇒ Country.
⇒ Grading system for recommendations.
⇒ Depression, any mention.
⇒ Depression screening and management, any recommendation.
⇒ CVD and depression interactions (drug–drug, disease–drug).
⇒ Level of evidence of depression recommendation.
⇒ Overall AGREE-II score of CPG.

AGREE-II, Appraisal of Guidelines for Research and Evaluation II; CAD, coronary artery disease; CPG, clinical practice guideline; CV, cardiovascular; CVD, cardiovascular disease; HF, heart failure; PAD, peripheral artery disease.

---

Research and Innovation (SERI) under contract number 22.00094, and from Wings Health in the context of a proof-of-concept study. GM is cofounder, member of the board and shareholder of Therayou AG, which is active in the field of digital and blended mental healthcare.

**Competing interests**  None declared.

**Patient and public involvement**  Patients and/or the public were not involved in the design, or conduct, or reporting, or dissemination plans of this research.

**Patient consent for publication**  Not applicable.

**Provenance and peer review**  Not commissioned; externally peer reviewed.

**ORCID iDs**
Dana Blatch Armon http://orcid.org/0000-0002-2160-9144
Aliki Buhayer http://orcid.org/0000-0002-6538-0273
Kevin Dobretz http://orcid.org/0000-0001-9138-4836
Gunther Meinlschmidt http://orcid.org/0000-0002-3488-193X
Edouard Battegay http://orcid.org/0000-0002-6538-0273

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
