## [Reviewer comments · BMJ Open]

ARTICLE DETAILS

TITLE (PROVISIONAL)	Clinical practice guidelines for cardiovascular disease: how is depression addressed? Protocol for a systematic review
AUTHORS	Blatch Armon, Dana; Buhayer, Alik; Dobretz, Kevin; Meinschmidt, Gunther; Bategay, Edouard

VERSION 1 – REVIEW

REVIEWER	Wilkowska, Alina Medical University of Gdansk, Psychiatry
REVIEW RETURNED	17-Feb-2023

GENERAL COMMENTS	This is a very timely and needed review since there is a great need for improvement in the care of CVD patients with depression. There is also lots of inconsistencies in this subject concerning the safety of antidepressants so I hope this matter can be discussed. I just have one remark. I do not really see how you can limit the review to primary care since most guidelines like European Heart Association do not have such a distinction, I do not really understand this idea. Besides that I think it will be an important piece of work in this field.
--

REVIEWER	Mizuno, Atsushi St. Luke's International Hospital
REVIEW RETURNED	23-Feb-2023

GENERAL COMMENTS	This protocol is for the systematic review about the guideline recommendation in patients with cardiovascular disease. This topic is important and this review will be useful both for clinicians and policy makers. I have several comments as below: Major 1. Although the main objective of this study to evaluate the guideline recommendation for patients with cardiovascular disease, the author describe some risks of cardiovascular disease in patients with depression in the introduction. Furthermore, in the PICAR statement, the population is limited in the primary care settings. The author should consider the inclusion and exclusion criteria carefully, because some primary care guidelines and guidelines for depression could include the management of depression in cardiovascular disease patients. Many guidelines for cardiovascular disease management were made for cardiologists not for primary care physicians. Please discuss.2. In data extraction, the two independent reviewers searched the terms "mental" and "psych". If they find the recommendation about some other mental disorders like anxiety and mood disorders, how
--

	to deal with them? If the author intended to include other potential disease subtypes, this protocol is not comprehensive and should be discussed carefully. 3. The cardiovascular disease includes broad spectrum of disease. In the search strategy, the author included hypertension as one of the cardiovascular disease categories. Did the author have any reasons to exclude other high-risk groups like aortic disease or diabetes? Minor 1. The reference of PICAR framework (P5, L27-28) seems different. As the author described, Johnstons article (Ref no.36) would be appropriate.
--	--

VERSION 1 – AUTHOR RESPONSE

Reviewer 1: Dr. Alina Wilkowska, Medical University of Gdansk

Comment: "This is a very timely and needed review since there is a great need for improvement in the care of CVD patients with depression. There is also lots of inconsistencies in this subject concerning the safety of antidepressants so I hope this matter can be discussed. I just have one remark. I do not really see how you can limit the review to primary care since most guidelines like European Heart Association do not have such a distinction, I do not really understand this idea. Besides that I think it will be an important piece of work in this field."

Response: We appreciate the reviewer's positive feedback. Regarding the comment regarding our intention to limit the review to the primary care setting, we acknowledge that some guidelines do not make a distinction between primary and specialty care. We have revised the manuscript accordingly, and we have now made it clear that the review will embrace primary as well as specialty (outpatient) care. However, we will maintain the exclusion of acute setting guidelines as our primary focus is on the management of depression in primary care and outpatient settings. (see P4, L 137-138: "1. Identify published CPGs on the management of CVD in the primary care and outpatient setting"; and also P7, 189 – 191, exclusion criteria: "1. CPGs for general outpatient management of adult patients with CVD (including specific guidelines for heart failure, coronary artery disease, stroke, peripheral arterial disease, aortic disease, diabetes, dyslipidemia and hypertension management) in the community setting.")

Reviewer 2: Dr. Atsushi Mizuno, St. Luke's International Hospital

Comment 1: "Although the main objective of this study to evaluate the guideline recommendation for patients with cardiovascular disease, the authors describe some risks of cardiovascular disease in patients with depression in the introduction. Furthermore, in the PICAR statement, the population is limited in the primary care settings. The author should consider the inclusion and exclusion criteria carefully, because some primary care guidelines and guidelines for depression could include the management of depression in cardiovascular disease patients. Many guidelines for cardiovascular disease management were made for cardiologists not for primary care physicians. Please discuss."

Response: We appreciate the reviewer's comment. We have revised the introduction to better reflect the aim of our study which is to focus on CVD guideline recommendations for the management of depression. In the revised version the examples of the risks of CVD in patients with depression were

removed. We agree that some guidelines may not distinguish between primary and specialty care, and we have clarified our inclusion and exclusion criteria accordingly, to include primary as well as specialty (outpatient) care guidelines (see P5, L162, Table 1: Population: Clinical indication: Management of patients with CVD in primary/specialty outpatient setting).

In this work we specifically decided to focus on CVD guidelines and not include depression guidelines. Though depression guidelines may include management recommendations for depression in CVD patients, our perspective in this work allows us to evaluate the current status in the cardiovascular field regarding mental health and specifically depression in these patients, and to evaluate what type and level of recommendations are available for treating physicians across the whole management process, including screening, when depression is not yet established. For this reason, we believe this current work can already be of benefit and hence important enough to be published on its own. A systematic review of depression guidelines and their recommendations for patients with CVD is indeed important and will be the object of our future work.

Comment 2: "In data extraction, the two independent reviewers searched the terms "mental" and "psych". If they find the recommendation about some other mental disorders like anxiety and mood disorders, how to deal with them? If the author intended to include other potential disease subtypes, this protocol is not comprehensive and should be discussed carefully."

Response: We appreciate the reviewer's comment and have added a section to the methods clarifying how we will handle recommendations for other mental disorders, such as anxiety and mood disorders. Using these somewhat general search terms will allow us to find any mention of depression or mental health-related topic within the text and then evaluate these sections for relevance. Although these terms are not intended to identify all mental health conditions as our focus is depression, all search results will be summarized and evaluated for inclusion in the results section of the review. (P7, L 204-207: "Once the final set of included CPGs has been obtained, DB and AB will conduct a search in the full text for any mention of depression, using the search terms "depress" "mental" "psych" and "mood".

These search terms will be used to identify all the places within the guidelines which mention the topic of mental health, in order to extract any information specifically regarding depression.")

Comment 3: "The cardiovascular disease includes broad spectrum of disease. In the search strategy, the author included hypertension as one of the cardiovascular disease categories. Did the author have any reasons to exclude other high-risk groups like aortic disease or diabetes?"

Response: We thank the reviewer for raising this important point. Other main high-risk groups such as dyslipidemia and diabetes deserve to be evaluated in the same way as hypertension. Accordingly, we have widened the search to include these CVD-related diseases that are managed mainly in the primary practice and outpatient setting. Aortic disease was also added to the search terms (P8, L228, Table 2: "CPG topic (CVD or specific CV-related conditions, including CAD, HF, stroke, PAD, aortic disease, hypertension, diabetes, dyslipidemia)". Also, the Supplementary file 2 containing the Pubmed Search Example was updated to include all search terms.

Comment 4: "The reference of PICAR framework (P5, L27-28) seems different. As the author described, Johnstons article (Ref no.36) would be appropriate."

Response: We thank the reviewer for this comment. The PICAR framework (ref. 43) is part of the PRISMA P reporting checklist (ref. 42). Both reporting tools were developed specifically for systematic reviews, and are recommended by Johnston et al (ref. 34) in their proposed methodology for a for a systematic review of clinical practice guidelines. We provide the references for PICAR and PRISMA-P in addition to the Johnston reference.

VERSION 2 – REVIEW

REVIEWER	Wilkowska, Alina Medical University of Gdansk, Psychiatry
REVIEW RETURNED	27-Mar-2023

GENERAL COMMENTS	The authors have improved the manuscript according to reviewer's suggestions.
---

REVIEWER	Mizuno, Atsushi St. Luke's International Hospital
REVIEW RETURNED	27-Mar-2023

GENERAL COMMENTS	I am pleased to see that the section I referenced has been appropriately amended. After reviewing the updated content, I do not have any further comments or suggestions to make.
---